# Orthorexia and Orthorexia Nervosa: A Comprehensive Examination of Prevalence, Risk Factors, Diagnosis, and Treatment

**DOI:** 10.3390/nu15173851

**Published:** 2023-09-03

**Authors:** Omer Horovitz, Marios Argyrides

**Affiliations:** 1The Physiology and Behavior Laboratory, Tel-Hai Academic College, 9977 North Districts, Qiryat Shemona 1220800, Israel; 2Psychology Department, Tel-Hai Academic College, 9977 North Districts, Qiryat Shemona 1220800, Israel; 3Psychology Department, Neapolis University Pafos, 2 Danais Avenue, Paphos 8042, Cyprus; m.argyrides.1@nup.ac.cy

**Keywords:** orthorexia, orthorexia nervosa, risk factors, diagnosis, treatment

## Abstract

Orthorexia nervosa is an emerging and controversial eating disorder characterized by an obsessive preoccupation with healthy eating and an extreme fixation on food purity. Despite growing public interest in orthorexia, its classification as a distinct eating disorder remains a subject of ongoing debate in the mental health community. This paper comprehensively reviews the current literature on orthorexia nervosa, exploring the prevalence rates, risk factors, diagnosis, and treatment options. The paper offers an overview of orthorexia and its historical context and explores the challenges and considerations in diagnosing orthorexia and orthorexia nervosa. Specifically, the distinction between “orthorexia” and “orthorexia nervosa” is a debated issue in eating disorder research due to a lack of clear diagnostic criteria, making it challenging to accurately differentiate between an obsession with healthy eating and a more severe form with potential distress and impairment. Given the absence of formal diagnostic criteria, developing valid and reliable assessment tools is crucial to accurately identify and treat individuals experiencing these disorders. The paper’s final section covers the existing treatment approaches for orthorexia nervosa. Overall, the paper highlights the complex and multifaceted nature of orthorexia nervosa. This review contributes to the ongoing discourse surrounding orthorexia and provides valuable insights for clinicians, researchers, and stakeholders in the mental health and eating disorders fields.

## 1. Introduction

Orthorexia and orthorexia nervosa are distinct but related concepts within disordered eating behaviors. Orthorexia can be defined as an obsessive and extreme fixation on consuming only “pure” and “healthy” foods, primarily focusing on the quality and cleanliness of the diet [1]. Individuals with orthorexia exhibit rigid dietary rules, often avoiding entire food groups and becoming increasingly preoccupied with their meals’ sourcing, preparation, and nutritional content [2]. On the other hand, orthorexia nervosa represents a more severe and clinically significant form of this obsession, characterized by intense anxiety, distress, and impairments in daily functioning resulting from extreme dietary restrictions [2]. Both conditions are expected to emphasize food purity and health, but orthorexia nervosa is distinguished by marked distress and dysfunction. Orthorexia nervosa has garnered increased attention due to its potential impact on mental and physical health, and the growing prevalence of this condition calls for a clearer understanding of its definition, diagnosis, and treatment. Thus, even though the distinction between orthorexia and orthorexia nervosa is addressed at different parts of the review article, this article focuses on orthorexia nervosa.

Disordered eating behaviors have emerged as a critical public health concern, drawing increasing attention from researchers, healthcare professionals, and policymakers. These behaviors encompass a broad spectrum of irregular eating patterns and attitudes towards food, with significant implications for physical, psychological, and social well-being. Of particular concern is the rise in the prevalence of eating disorders such as anorexia nervosa, bulimia nervosa, and binge eating disorder, which pose significant health risks and can be life-threatening if left untreated [3,4]. Moreover, subclinical disordered eating, including orthorexia and other unhealthy preoccupations with food and body image, has also increased alarmingly. The media’s portrayal of unrealistic body standards, the pervasive influence of social media, and societal pressures to conform to particular body ideals contribute to the escalation of disordered eating behaviors among diverse age groups and genders [5]. Consequently, these rising concerns necessitate a comprehensive examination of the factors contributing to the development and perpetuation of these behaviors to inform effective prevention strategies, early intervention programs, and evidence-based treatment approaches.

In recent years, research has increasingly shed light on the detrimental consequences of disordered eating behaviors on individuals’ physical and mental health. Prolonged engagement in restrictive or excessive eating patterns can lead to severe nutritional deficiencies, electrolyte imbalances, and disturbances in metabolic functions, endangering the individual’s overall health [6]. Furthermore, disordered eating is intricately linked to the development of various mental health conditions, including depression, anxiety, and body dysmorphic disorder [7,8]. These psychological repercussions are exacerbated by the profound impact of societal stigma and misconceptions surrounding eating disorders, which can hinder affected individuals from seeking timely and appropriate treatment. Additionally, disordered eating behaviors often result in significant impairments in social and occupational functioning, exacerbating feelings of isolation and perpetuating the disorder cycle [9]. Given the multifaceted nature of these concerns, a comprehensive understanding of disordered eating behaviors is indispensable for healthcare professionals, policymakers, and educators to collaboratively address the rising prevalence and harmful consequences of these conditions, ultimately striving to promote healthier relationships with food and body image in society.

There has been a noticeable upsurge in the prevalence of “clean eating” ideals within society [2]. This trend reflects a growing societal emphasis on pursuing healthier lifestyles and making more mindful dietary choices. “Clean eating” generally entails prioritizing whole, unprocessed foods while limiting or avoiding heavily refined and artificial ingredients. This movement has fueled a desire for improved well-being, weight management, and increased energy levels [10]. However, discussions around clean eating also come with debates about the potential for creating rigid eating patterns, promoting unrealistic body standards, and contributing to the stigmatization of certain foods [2]. As this trend continues to evolve, striking a balance between informed nutritional choices and avoiding extreme dietary restrictions remains an ongoing conversation.

Orthorexia and orthorexia nervosa have garnered increased attention in the field of disordered eating behaviors. Prevalence studies indicate that though the exact rates remain uncertain, both conditions are becoming more prevalent, especially in Western societies with a strong emphasis on health and nutrition [11]. Risk factors associated with these disorders include personality traits such as perfectionism and neuroticism, a history of dieting, body dissatisfaction, and exposure to media promoting unrealistic body ideals and “clean eating” trends [12]. The diagnosis and conceptualization of orthorexia and orthorexia nervosa remain a subject of much debate. Orthorexia nervosa is not considered an official mental disorder; it is not listed by the Diagnostic and Statistical Manual of Mental Disorders (DSM-5 [13]) nor by the International Classification of Diseases (ICD-10 [14]). However, the proposed diagnostic tools and questionnaires aid in identifying individuals with disordered eating habits related to food purity and health obsessions. Treatment approaches involve a multidisciplinary approach, including psychoeducation, cognitive-behavioral therapy (CBT), nutritional counseling, and mindfulness-based interventions. Nevertheless, further research is warranted to better understand the prevalence, underlying risk factors, and effective treatment strategies for these emerging eating disorders.

## 2. Historical Context and Emergence of Orthorexia and Orthorexia Nervosa

The historical context and emergence of orthorexia and orthorexia nervosa can be traced back to the late 20th and early 21st centuries, coinciding with a growing societal emphasis on health, nutrition, and wellness. The term “orthorexia” was first coined in 1997 by Dr. Steven Bratman, a holistic medical practitioner, who observed a pattern of excessive preoccupation with healthy eating among his patients [15]. The concept gained initial recognition as an informal term to describe an obsession with pure and healthy foods. Over time, as awareness of disordered eating patterns increased, researchers and healthcare professionals recognized orthorexia as a distinct phenomenon deserving further investigation. Subsequently, the term “orthorexia nervosa” emerged to describe a more severe variant characterized by intense anxiety, distress, and functional impairments related to the rigid pursuit of an idealized and “pure” diet [1]. The rise of social media and the internet facilitated the dissemination of dietary trends, fostering a culture that glorified specific eating patterns and demonized others. The relentless pursuit of “clean eating” and the proliferation of unscientific health claims may have contributed to the amplification of orthorexic behaviors [16]. The historical context surrounding these disorders highlights the complex interplay between societal norms, technological advancements, and the human desire for health and wellness, emphasizing further research and understanding to address the emerging challenges of orthorexia and orthorexia nervosa.

The distinction between orthorexia nervosa and other eating disorders lies in the primary focus and underlying motivations behind the disordered eating behaviors. Whereas eating disorders like anorexia nervosa, bulimia nervosa, and binge eating disorder are primarily characterized by disturbances in food intake quantity or patterns, orthorexia nervosa centers around the quality and purity of the consumed food [17]. Individuals with orthorexia nervosa display an excessive preoccupation with consuming only “healthy” or “pure” foods, often leading to the strict elimination of entire food groups, rigid dietary rules, and an intense fixation on the nutritional content of their meals [2]. In contrast, anorexia nervosa involves severe restriction of caloric intake, often leading to significantly low body weight [18]. Further, bulimia nervosa is characterized by recurrent episodes of binge eating followed by compensatory behaviors such as purging or excessive exercise [18]. Recurrent episodes of excessive food consumption characterize binge eating disorder without compensatory behaviors [18]. Although orthorexia nervosa shares some features with other eating disorders, it is distinguishable by its emphasis on the perceived health benefits of food choices and the absence of concerns about body shape or weight. Recognizing these distinctions is crucial for accurate diagnosis, treatment planning, and providing appropriate support for individuals struggling with disordered eating patterns.

Assessing the prevalence and incidence rates of orthorexia nervosa is challenging due to the relatively recent recognition of these conditions and the absence of standardized diagnostic criteria. As a result, research on their prevalence remains limited and inconclusive. However, available studies suggest a rising concern surrounding these disorders, particularly in Western societies where health and wellness trends have become increasingly popular [19]. Prevalence estimates for orthorexia vary widely, with some studies reporting rates as high as 90.6% among specific populations [11,20,21]. It is important to note that the lack of consensus on diagnostic criteria and assessment tools may contribute to the variability in prevalence rates across different studies. Furthermore, orthorexia nervosa, a more severe and clinically significant form, is believed to be less prevalent than orthorexia but may share similar risk factors and psychological underpinnings [11]. As the understanding of orthorexia and orthorexia nervosa continues to evolve, further research employing standardized diagnostic criteria and rigorous methodologies is needed to obtain more accurate and comprehensive prevalence and incidence data for these emerging eating disorders.

## 3. Risk Factors and Associated Psychological and Sociocultural Influences

Risk factors associated with orthorexia and orthorexia nervosa encompass a complex interplay of psychological and sociocultural influences. At the individual level, certain personality traits have been linked to an increased vulnerability to developing these disorders [22]. Perfectionism, high levels of neuroticism, and a tendency towards obsessive-compulsive traits may predispose individuals to engage in rigid and extreme dietary behaviors [22]. Furthermore, a history of dieting and body dissatisfaction can contribute to developing disordered eating patterns centered around food purity and health obsession [23,24]. At the sociocultural level, the media’s portrayal of idealized body images and “clean eating” trends can pressure individuals to conform to unrealistic health and nutrition standards [25]. Social media platforms, in particular, play a pivotal role in disseminating these trends, promoting harmful dietary practices, and fostering a sense of social comparison [21]. Additionally, the widespread availability of misinformation and unscientific health claims can further fuel fears about “toxic” foods, perpetuating the pursuit of restrictive and narrow dietary choices. Societal norms prioritizing physical appearance and equating health with dietary purity may inadvertently encourage the emergence and maintenance of orthorexia and orthorexia nervosa, highlighting the need for targeted prevention and intervention efforts to address these risk factors [26].

The psychological and sociocultural influences on orthorexia and orthorexia nervosa are closely intertwined, creating a complex web of factors contributing to these disorders’ development and perpetuation. At the psychological level, individuals who experience high stress, anxiety, or dissatisfaction with other aspects of their lives may turn to food and the illusion of dietary control to cope or gain a sense of mastery [27,28]. The obsession with consuming only “pure” and “clean” foods has ignited discussions about the intersection of personal illness fears and extreme dietary habits. Individuals with orthorexia often feel overwhelming anxiety about consuming anything perceived as unhealthy, so their fixation on food quality can severely impact their physical and mental well-being [1]. This highlights the complex interplay between a genuine concern for health and the development of an unhealthy preoccupation with food choices. Though originating from genuine concern, the fear of falling ill can morph into a consuming obsession that requires careful consideration within the broader context of mental health and balanced nutrition.

Moreover, pursuing “clean eating” and rigid dietary rules can provide a sense of identity and purpose, offering a way to define oneself within a health-conscious community [10]. Sociocultural influences, on the other end, play a significant role in shaping societal norms and attitudes toward food, body image, and health. Glorifying certain dietary practices and demonizing others in media and popular culture can foster an all-or-nothing approach to food consumption, reinforcing the belief that only “pure” and “healthy” choices are acceptable [10]. Societal pressures to conform to these standards and the fear of being judged or ostracized for deviating from them may escalate orthorexic behaviors. By understanding the intertwined psychological and sociocultural influences, healthcare professionals and policymakers can develop more effective strategies to address these risk factors and promote a healthier and more balanced relationship with food and nutrition.

## 4. Definition and Diagnostic Criteria

Orthorexia and orthorexia nervosa have gained recognition as emerging eating disorders, so efforts have been made to propose diagnostic criteria to facilitate their identification and assessment. However, it is essential to note that these disorders are not officially recognized in the DSM-5 or the ICD-10 [13,14]. Nevertheless, several researchers and clinicians have proposed diagnostic criteria based on the observed characteristics of individuals with these conditions [1,16,29]. Proposed criteria for orthorexia nervosa typically include features such as an intense preoccupation with the quality and purity of food, an excessive focus on the nutritional content of meals, rigid dietary rules and restrictions, and a significant impact on daily life and functioning due to these behaviors. For orthorexia nervosa, in addition to the above features, the proposed diagnostic criteria may involve severe anxiety and distress related to deviations from strict dietary standards, resulting in psychological impairment and significant disruptions in social, occupational, or educational activities [29]. Though these proposed criteria offer valuable insights into the clinical presentation of orthorexia and orthorexia nervosa, further research and consensus-building efforts are necessary to establish standardized and validated diagnostic criteria to improve diagnostic accuracy and treatment approaches and ultimately advance understanding these emerging eating disorders. Donini et al. (2022) have made a very recent attempt to reach a consensus on the directions that need to be taken on this issue [30].

Diagnosing orthorexia nervosa poses several challenges due to its relatively recent recognition and the absence of standardized diagnostic criteria in major psychiatric classification systems. One of the primary challenges lies in differentiating orthorexia nervosa from other eating disorders. The symptoms and behaviors associated with orthorexia nervosa may overlap with those of anorexia nervosa, as both disorders involve rigid dietary rules and restrictions [17]. However, unlike anorexia nervosa, orthorexia nervosa does not necessarily involve an intense fear of gaining weight or body image distortion. Additionally, distinguishing orthorexia nervosa from other forms of disordered eating, such as obsessive-compulsive disorder (OCD) and generalized anxiety disorder, is complicated by the overlap in symptoms related to excessive preoccupation with health and cleanliness [2].

Furthermore, the proposed diagnostic criteria for orthorexia nervosa are not yet universally accepted, leading to inconsistencies in assessment and diagnosis among clinicians and researchers. The lack of standardized criteria also hampers the accurate estimation of the prevalence and incidence rates of orthorexia nervosa, impeding our understanding of this emerging eating disorder’s full extent and impact. Addressing these diagnostic challenges and establishing clear distinctions between orthorexia nervosa and other eating disorders is essential to improving clinical recognition, early intervention, and evidence-based treatment approaches for individuals struggling with these complex eating behaviors. A recent attempt has been made to reach a consensus on the conceptualization and diagnosis of orthorexia nervosa [30]. The consensus provides a new and more precise direction to the field that requires further research to support the written document. Additionally, the literature has attempted to distinguish between healthy orthorexia and orthorexia nervosa [31,32,33]. This distinction is essential in avoiding over-diagnosis and stigmatizing all healthy eating (including vegans and vegetarians) [30].

For a concise summary of this topic, please see Table 1.

## 5. Psychological and Physical Consequences

The psychological impact of orthorexia and orthorexia nervosa on mental health is profound and multifaceted. Individuals grappling with these disorders often experience significant distress and anxiety surrounding their food choices, leading to an all-encompassing preoccupation with dietary purity and health [2]. The relentless pursuit of “clean eating” can lead to social isolation and impaired functioning in various domains of life as the focus on food dominates these individuals’ thoughts and behaviors. Feelings of guilt, shame, and failure may arise when individuals deviate from their self-imposed dietary rules, exacerbating anxiety and reinforcing the cycle of obsessive thoughts and compulsive behaviors [34]. Moreover, individuals with orthorexia and orthorexia nervosa may experience diminished enjoyment in social gatherings or communal meals due to the stress and anxiety related to food choices, leading to social withdrawal and feelings of alienation. As these disorders progress, individuals may develop distorted beliefs about the relationship between diet and health, leading to rigid thinking and impaired cognitive flexibility [35]. Ultimately, the psychological toll of orthorexia and orthorexia nervosa can significantly impact the affected individual’s emotional well-being, interpersonal relationships, and overall quality of life, underscoring the need for early recognition and intervention to address these complex and debilitating eating disorders [36].

Orthorexia, characterized by an obsessive fixation on consuming only “pure” and “healthy” foods, can lead to various nutritional deficiencies and health complications. Individuals with orthorexia often follow restrictive diets, eliminating entire food groups or severely limiting their food choices based on self-imposed criteria for purity and healthiness. As a result, they may not obtain a well-balanced and varied diet, leading to deficiencies in essential nutrients such as proteins, fats, vitamins, and minerals [37]. Prolonged adherence to such restrictive diets can compromise nutritional status, leading to physical health issues like malnutrition, osteoporosis, and weakened immune function [38,39]. Furthermore, individuals with orthorexia may become fixated on consuming only raw or unprocessed foods, which can increase the risk of foodborne illnesses and infections. The fear of consuming “impure” foods can also lead to heightened anxiety and stress surrounding eating, further exacerbating the psychological impact of orthorexia [40]. It is crucial for healthcare professionals to be aware of the potential nutritional consequences of orthorexia and to provide appropriate support and intervention to address both the physical and psychological aspects of the disorder. Early recognition and intervention are vital to mitigate the risk of long-term health complications and promote a balanced and sustainable approach to nutrition and overall well-being.

The impact of orthorexia and orthorexia nervosa on quality of life and social functioning is considerable, encompassing various aspects of an individual’s daily existence. The relentless preoccupation with food purity and rigid dietary rules can dominate one’s thoughts and behaviors, leading to a diminished capacity to engage in social activities and enjoy communal meals [26]. As individuals with these disorders prioritize their health-focused dietary restrictions, they may experience heightened anxiety and stress in social situations that involve food, leading to avoidance of social gatherings or events centered around meals [41]. The resultant social isolation and withdrawal can negatively affect interpersonal relationships and impede the development of meaningful connections with others. The obsession with “clean eating” can also extend to family members or friends, creating tension and strain in relationships when others fail to adhere to the same dietary principles. The profound impact of orthorexia and orthorexia nervosa on social functioning can contribute to feelings of loneliness, depression, and a reduced sense of well-being [36]. As the disorders progress, individuals may increasingly prioritize their dietary rules over personal relationships and professional pursuits, compromising their overall quality of life. Understanding the impact of these eating disorders on social functioning is crucial in devising effective interventions that address both the psychological and social aspects of orthorexia and orthorexia nervosa to enhance the overall well-being and social connectedness of affected individuals.

Impaired social functioning is a salient aspect of orthorexia nervosa [32,35]. As individuals with orthorexia become increasingly preoccupied with adhering to rigid dietary rules, their social interactions often suffer [1]. This is primarily due to the restrictive nature of their eating habits, which can lead to avoidance of social gatherings involving food, dining out, or participating in shared meals [33]. The intense anxiety surrounding food quality and composition may distance sufferers from friends and family who do not share their dietary restrictions, resulting in isolation and alienation [2,12,26]. The rigid mindset and time-consuming behaviors associated with orthorexia can also hinder the ability to engage in spontaneous social activities, further compromising their social connections. Ultimately, impaired social functioning underscores the need for a comprehensive approach to address both the physical and psychological aspects of orthorexia nervosa to help individuals regain a healthy relationship with food and interpersonal interactions.

## 6. Personality and Sociocultural Influences

Psychological factors are central to developing and perpetuating orthorexia and orthorexia nervosa. Individuals at risk of these disorders often exhibit personality traits such as perfectionism, obsessive-compulsive tendencies, and high neuroticism [22]. The desire for control and the need for orderliness may manifest in the strict adherence to self-imposed dietary rules and an excessive fixation on food purity and healthiness [22]. Moreover, individuals with low self-esteem or body dissatisfaction may seek a sense of identity and self-worth by pursuing an idealized “healthy” lifestyle, using food choices to gain a sense of accomplishment and validation [5]. Furthermore, underlying anxiety or a fear of adverse health consequences may drive individuals to adopt increasingly restrictive diets to protect their well-being, inadvertently exacerbating their obsessive behaviors [42]. Additionally, the influence of stress and life transitions may trigger the onset or escalation of these disorders, as the perceived need for control over one’s diet and health may intensify during times of uncertainty [43]. Understanding these psychological factors is critical for the early recognition, prevention, and treatment of orthorexia and orthorexia nervosa, enabling tailored interventions that address the underlying motivations and cognitive patterns driving these complex and challenging eating disorders.

Sociocultural influences and media significantly shape disordered eating behaviors, including orthorexia and orthorexia nervosa. Modern societies strongly emphasize appearance, equating health and beauty with thinness and a particular body ideal. Media portrayals of unrealistic and often digitally enhanced body images further reinforce these societal norms, perpetuating a narrow and unattainable standard of beauty [44]. As a result, individuals, particularly adolescents and young adults, may internalize these ideals and engage in extreme dieting or disordered eating behaviors to conform to perceived expectations [45]. Social media platforms, in particular, have become powerful conduits for disseminating dietary trends and promoting a “clean eating” culture. Constant exposure to curated images of seemingly “perfect” bodies and “healthy” eating habits can create an environment that normalizes extreme dieting and reinforces that adhering to strict dietary rules is the key to achieving health and success [5,25,44]. To address the impact of sociocultural influences and media on disordered eating behaviors, public health campaigns and education initiatives are needed to promote body positivity, challenge unrealistic beauty ideals, and foster a balanced and evidence-based approach to nutrition and overall well-being.

Personality traits and obsessive-compulsive tendencies play a significant role in developing and maintaining various psychological conditions, including disordered eating behaviors like orthorexia and orthorexia nervosa. Research suggests that individuals with these disorders often exhibit specific personality traits contributing to their susceptibility [22,23,41]. Perfectionism is a prominent personality trait associated with orthorexia and orthorexia nervosa, as individuals may seek to achieve an idealized and flawless standard of health and nutrition [23]. This drive for perfection can manifest in rigid dietary rules, strict adherence to self-imposed restrictions, and an unrelenting pursuit of an unattainable dietary ideal. Moreover, obsessive-compulsive tendencies can intensify the fixation on food purity and healthiness, leading to repetitive thoughts and behaviors related to dietary choices [17]. The need for control and orderliness, characteristic of obsessive-compulsive traits, can drive individuals to meticulously plan and scrutinize their meals, resulting in significant time and mental energy spent on food-related activities. The combination of perfectionism and obsessive-compulsive tendencies creates a fertile ground for developing orthorexia and orthorexia nervosa, underscoring the importance of addressing these personality traits in therapeutic interventions to promote a more balanced and flexible approach to food and nutrition.

## 7. Assessment and Diagnosis

Identifying and assessing orthorexia and orthorexia nervosa pose several challenges, primarily due to the absence of standardized diagnostic criteria in major psychiatric classification systems such as the DSM-5 and ICD-10 [13,14]. As a relatively new and evolving concept, the lack of universally accepted criteria makes establishing clear clinical recognition and diagnosis guidelines difficult. Additionally, the boundaries between orthorexia, other eating disorders, and mental health conditions with similar symptoms, such as obsessive-compulsive disorder (OCD) or generalized anxiety disorder, can be blurry, further complicating the diagnostic process [17,29,36,42]. The overlap in symptoms and behaviors among these disorders requires a comprehensive assessment that considers the context and underlying motivations of the individual’s disordered eating patterns. Moreover, the reliance on self-report measures and subjective assessments introduces potential biases and challenges in accurately gauging the severity and impact of orthorexia and orthorexia nervosa on an individual’s well-being. Standardized and validated diagnostic tools tailored to these disorders are still in the early stages of development, highlighting the need for further research and collaboration among experts to establish a clear and comprehensive framework for identifying and assessing orthorexia and orthorexia nervosa, ultimately improving diagnostic accuracy and guiding appropriate treatment interventions.

Efforts to assess orthorexia nervosa have led to the development of specific questionnaires and clinical tools designed to aid in its identification and evaluation. The “Bratman Orthorexia Test” (BOT) is among the widely used assessment instruments [46]. The BOT consists of 10 items that assess an individual’s preoccupation with healthy eating, the impact of this behavior on daily life, and anxiety and guilt associated with deviations from self-imposed dietary rules. Another notable tool is the “Dusseldorf Orthorexia Scale” (DOS), which measures orthorexic behaviors and attitudes on a 10-point Likert scale [47]. The “Teruel Orthorexia Scale” (TOS) also assesses obsessive food-related thoughts and the time spent on dietary habits [48]. Clinical interviews conducted by trained professionals further contribute to a comprehensive assessment, allowing for a deeper understanding of the individual’s experiences and motivations related to orthorexia nervosa. Though these questionnaires and clinical tools provide valuable insights into the presence and severity of orthorexic behaviors, ongoing research is needed to refine and validate these assessment measures, ensuring their reliability and effectiveness in accurately identifying orthorexia nervosa and guiding appropriate treatment strategies.

The differential diagnosis of orthorexia nervosa involves careful consideration of the overlapping symptoms with other eating disorders and mental health conditions. Orthorexia nervosa shares similarities with anorexia nervosa, particularly regarding restrictive eating patterns and food-related preoccupation [49]. Distinguishing between the two disorders relies on the absence of body image distortion and the primary focus on food quality and purity in orthorexia nervosa [29]. Additionally, differentiating orthorexia nervosa from other eating disorders like bulimia nervosa and binge eating disorder entails thoroughly evaluating compensatory behaviors, such as purging or binge eating, typically absent in the orthorexia [50]. Comorbidity is also pertinent, as individuals with orthorexia nervosa may experience co-occurring mental health conditions such as generalized anxiety disorder, obsessive-compulsive disorder (OCD), or depression [2,12,16,26,29,36,50]. These comorbidities can further complicate the diagnostic process and require a comprehensive assessment to differentiate the primary focus and impact of orthorexia nervosa on an individual’s well-being. Accurate differential diagnosis and assessment of comorbidities are crucial for effective treatment planning, as addressing the specific aspects of orthorexia nervosa and the co-existing mental health conditions contributes to comprehensive and tailored interventions for individuals struggling with these complex eating behaviors.

Yet again, the differentiation between “orthorexia” and “orthorexia nervosa” remains a complex and debated issue within the field of eating disorders. The lack of a clear distinction in existing diagnostic tools poses challenges for accurate assessment and treatment. Whereas “orthorexia” generally refers to an obsession with healthy eating and a fixation on consuming pure and uncontaminated foods [15], “orthorexia nervosa” suggests a more severe form involving distress, impairment, and potential physical or psychological consequences [1]. However, the absence of standardized diagnostic criteria for either condition makes it difficult to differentiate between them in practice. The concept of orthorexia nervosa is not yet formally recognized by major diagnostic manuals like the DSM-5, further complicating matters. Although clinicians and researchers are working to refine and establish clear criteria to identify and address these conditions effectively, the lack of consensus remains a significant hurdle in accurately distinguishing between them.

For a concise summary of this topic, please see Table 2 and Table 3.

## 8. Treatment Approaches

Psychoeducation and cognitive-behavioral therapy (CBT) are prominent treatment approaches for individuals with orthorexia nervosa [2]. Psychoeducation plays a foundational role in treatment, providing individuals with information about the nature of orthorexia nervosa, its potential consequences on physical and mental health, and the underlying psychological factors contributing to the disorder [2]. This knowledge empowers individuals to recognize and challenge distorted beliefs and cognitive patterns surrounding food and health, fostering a deeper understanding of their disordered eating behaviors. Conversely, CBT offers a structured, evidence-based intervention that targets maladaptive thought processes and behaviors related to orthorexia nervosa. Through CBT, individuals learn to identify and reframe rigid and unhealthy thoughts about food, nutrition, and body image, and they acquire coping strategies to manage anxiety and distress related to dietary deviations.

Additionally, CBT assists in developing a more balanced and flexible approach to eating, helping individuals gradually reintegrate previously avoided foods into their diet [51]. Regular monitoring of dietary patterns and emotional responses is a crucial component of CBT, facilitating a greater awareness of triggers and stressors contributing to “orthorexic” behaviors. Combining psychoeducation and CBT equips individuals with essential skills and tools to challenge and modify maladaptive behaviors and beliefs, fostering positive changes in dietary habits and promoting overall psychological well-being in treating orthorexia nervosa.

Nutritional counseling and intuitive eating approaches are valuable components in the comprehensive treatment of orthorexia nervosa. Nutritional counseling involves working with a registered dietitian or nutritionist specializing in eating disorders to provide personalized guidance and support in restoring a healthy and balanced relationship with food [52]. The focus of nutritional counseling for individuals with orthorexia nervosa is to address nutritional deficiencies and imbalances resulting from restrictive eating patterns while promoting a more flexible and inclusive approach to food choices. The dietitian collaborates with the individual to develop a structured meal plan that meets their nutritional needs, considering their food preferences and dietary restrictions and encouraging variety and moderation. In contrast, intuitive eating emphasizes attunement to internal hunger and satiety cues, enabling individuals to reconnect with their body’s natural signals for nourishment. Intuitive eating helps individuals overcome rigid dietary rules and adopt a more mindful and non-judgmental approach to eating, promoting self-compassion and body acceptance [53]. By integrating nutritional counseling and intuitive eating approaches, individuals with orthorexia nervosa can gradually learn to trust their body’s signals, develop a more positive relationship with food, and cultivate a sustainable and balanced eating method that supports their physical and psychological well-being. Additionally, there is a vast body of research highlighting the benefits of promoting positive body image in treating eating disorders [54].

Mindfulness and acceptance-based treatments are emerging as promising therapeutic approaches for individuals with orthorexia nervosa [55]. Mindfulness practices involve cultivating present moment awareness and non-judgmental acceptance of one’s thoughts, emotions, and bodily sensations. This approach allows individuals to observe their obsessive thoughts and anxiety about food without reacting to them impulsively, promoting a sense of detachment from these distressing experiences. Individuals can develop greater insight into the underlying triggers and motivations driving their disordered eating behaviors by increasing awareness of their thoughts and emotions. Acceptance-based strategies complement mindfulness practices, encouraging individuals to embrace their vulnerabilities and imperfections without self-criticism or judgment. This approach fosters self-compassion and reduces the need for dietary perfectionism, thus challenging the core beliefs associated with orthorexia nervosa. Integrating mindfulness and acceptance into treatment helps individuals cultivate a more balanced and flexible relationship with food, promoting psychological resilience and overall well-being. Individuals can embark on a healing journey by developing greater self-awareness and self-compassion, gradually overcoming the grip of orthorexia nervosa and cultivating a healthier and more fulfilling life [56]. Further research is needed to explore the efficacy and specific components of mindfulness and acceptance-based treatments for orthorexia nervosa to guide evidence-based interventions and optimize treatment outcomes for individuals struggling with this complex eating disorder.

Family and social support are instrumental in recovery for individuals with orthorexia nervosa. The involvement of family members and close friends can significantly influence treatment adherence and outcomes [57]. By providing a nurturing and understanding environment, family support can reduce feelings of isolation and stigma, creating a safe space for individuals to share their struggles and challenges during recovery. Family therapy can facilitate open communication and address any dysfunctional family dynamics that may have contributed to the development of orthorexia nervosa. It also offers an opportunity for family members to learn about the disorder and how they can play a positive role in supporting the individual’s recovery. Beyond the family context, social support from peers and support groups can be invaluable [58]. Connecting with others who have experienced similar challenges can foster empathy, a sense of belonging, and shared growth. Support groups or group therapy sessions allow individuals to share experiences, offer encouragement, and exchange coping strategies. Ultimately, the combined impact of family and social support can create a robust support network that bolsters an individual’s resilience and commitment to recovery, providing a solid foundation for sustained progress and improved overall well-being.

For a concise summary of this topic, please see Table 4.

## 9. Future Directions and Research Needs

The need for longitudinal studies on the course and outcomes of orthorexia and orthorexia nervosa is paramount to enhance our understanding of these emerging eating disorders and inform evidence-based interventions. Currently, the literature on orthorexia and orthorexia nervosa is limited by cross-sectional studies, which offer a snapshot of the disorders at a particular time. Longitudinal studies, on the other hand, can provide critical insights into the natural trajectory of these disorders over an extended period, shedding light on their stability, progression, and potential remission. By following individuals with orthorexia and orthorexia nervosa over time, researchers can explore the factors that influence the development and exacerbation of these disorders and identify protective factors that contribute to recovery and improved outcomes. Additionally, longitudinal studies can assess the impact of interventions and treatment approaches, examining their effectiveness in promoting recovery and preventing relapse. Long-term follow-up data are essential to evaluate the long-term consequences of orthorexia and orthorexia nervosa on physical and mental health, including potential nutritional deficiencies, psychological distress, and social functioning. Overall, longitudinal studies play a pivotal role in deepening our understanding of these complex eating disorders, guiding the development of targeted interventions, and ultimately improving the quality of care and support for individuals affected by orthorexia and orthorexia nervosa.

The need for cross-cultural studies and a global understanding of the prevalence of orthorexia nervosa is essential to recognize the impact of cultural influences on the development and manifestation of this eating disorder. The concept of “healthy” eating and pursuing dietary purity may vary significantly across cultures, affecting the presentation and recognition of orthorexia nervosa in different populations. Cross-cultural studies can explore how cultural norms, beliefs, and societal pressures influence attitudes toward food, body image, and health, potentially shedding light on unique risk factors and protective factors specific to different cultural contexts. Moreover, understanding the global prevalence of orthorexia nervosa is crucial to address its impact on public health. Given the widespread dissemination of health and wellness trends through digital media and the internet, orthorexia nervosa will likely transcend geographical boundaries. By examining the prevalence of this disorder across diverse regions, researchers and policymakers can effectively identify high-risk populations and allocate resources for prevention and early intervention efforts. Ultimately, the global understanding of orthorexia nervosa through cross-cultural studies will contribute to developing culturally sensitive diagnostic criteria, treatment approaches, and public health strategies, advancing the global recognition and management of this complex eating disorder.

Developing a consensus on diagnostic criteria and assessment tools for orthorexia nervosa is critical to advance research, improve clinical recognition, and enhance treatment outcomes for individuals with this emerging eating disorder. Currently, the lack of standardized diagnostic criteria poses challenges in accurately identifying and diagnosing orthorexia nervosa, leading to variability in prevalence estimates and hindering comparability across studies. The absence of agreed-upon assessment tools further complicates the diagnostic process, making it challenging to capture the full spectrum and severity of orthorexic behaviors and their impact on individuals’ well-being. Developing a comprehensive and validated set of diagnostic criteria and assessment measures is crucial to differentiate orthorexia nervosa from other eating disorders and related mental health conditions, enabling tailored treatment approaches that address the specific needs and challenges of those struggling with this complex disorder. A consensus on diagnostic criteria and assessment tools will enhance the accuracy of diagnosis and promote early recognition and intervention, ultimately improving outcomes and providing a solid foundation for advancing research and understanding of orthorexia nervosa.

## 10. Conclusions

This review paper comprehensively examines orthorexia and orthorexia nervosa, encompassing prevalence, risk factors, diagnosis, and treatment strategies. Reviewing existing literature and empirical data, the paper sheds light on the growing concerns surrounding these emerging eating disorders. It highlights the complexities in diagnosing orthorexia and orthorexia nervosa due to the absence of standardized criteria and the need for further research in developing accurate assessment tools. Even though a recent consensus paper by Donini et al. [30] has made good progress towards a better understanding of the diagnosis and conceptualization of orthorexia nervosa, the establishment of unified diagnostic criteria is urgently needed to facilitate accurate identification, research, and effective treatment of this complex eating disorder. The paper also identifies personality traits, sociocultural influences, and psychological factors as significant risk factors contributing to the development of these disorders. Sharing and accumulating the known data should better define these disorders and provide valuable insights for healthcare professionals, researchers, and policymakers in addressing the multifaceted challenges associated with orthorexia and orthorexia nervosa.

## Figures and Tables

**Table 1 nutrients-15-03851-t001:** Challenges in identifying and assessing orthorexia.

Lack of standardized diagnostic criteria in major psychiatric classification systems (DSM-5, ICD-10).
Absence of universally accepted criteria for clear clinical recognition and diagnosis.
Blurred boundaries between orthorexia, other eating disorders, and similar mental health conditions (OCD, anxiety disorder).
Overlapping symptoms necessitate comprehensive assessment considering context and motivations.
Reliance on self-report measures introduces potential biases and challenges.
Need for standardized and validated diagnostic tools tailored to orthorexia and orthorexia nervosa.
Ongoing research and collaboration among experts are essential for a comprehensive framework.

**Table 2 nutrients-15-03851-t002:** Common assessment tools for orthorexia nervosa.

“Bratman Orthorexia Test” (BOT): 10 items qualitatively assessing preoccupation with healthy eating, impact on daily life, anxiety, and guilt.
“ORTHO-15”: 15-item self-report measure assessing perceptions of eating healthy food, attitudes governing food selection, food consumption habits, and the influence of food in an individual’s daily life. “Dusseldorf Orthorexia Scale” (DOS): 10-item measure addressing orthorexic behaviors and attitudes.
“Teruel Orthorexia Scale” (TOS): 17-item self-report measure assessing “healthy orthorexia”, indicating a healthy, non-pathological interest in proper nutrition, and “orthorexia nervosa”, representing an extreme preoccupation with healthy diet leading to emotional, social, and cognitive impairments.
Clinical interviews by professionals contribute to a deeper understanding.
Ongoing research needed to refine and validate assessment measures.

**Table 3 nutrients-15-03851-t003:** Differential diagnosis and comorbidity.

Orthorexia nervosa vs. anorexia nervosa: focus on food quality and purity in orthorexia, without body image distortion.
Orthorexia nervosa vs. bulimia nervosa/binge eating disorder: evaluate compensatory behaviors.
Comorbidity with mental health conditions (anxiety disorder, OCD, depression).
Comprehensive assessment required for accurate differential diagnosis and treatment planning.

**Table 4 nutrients-15-03851-t004:** Treatment approaches for orthorexia nervosa.

*Psychoeducation and CBT*
Psychoeducation informs about orthorexia, its consequences, and psychological factors.
CBT targets maladaptive thoughts and behaviors; reframing beliefs, coping with anxiety.
Encourages reintegration of avoided foods, monitoring triggers.
*Nutritional Counseling and Intuitive Eating*
Nutritional counseling restores balance and considers preferences and restrictions.
Intuitive eating focuses on internal cues, mindfulness, and body acceptance.
The combination helps the individual to develop a positive relationship with food.
*Mindfulness and Acceptance-based Treatment*
Mindfulness enhances awareness of thoughts and emotions, aiding with impulsivity.
Acceptance fosters self-compassion and reduces perfectionism.
Treatment aims for balanced food relationship and psychological well-being.
*Family and Social Support*
Family involvement reduces isolation and addresses dynamics.
Family therapy promotes understanding and support.
Peer support groups foster empathy and coping strategies.
Family and social support enhance recovery and overall well-being.

## Data Availability

Not applicable.

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
