# Peer review of "Orthorexia and Orthorexia Nervosa: A Comprehensive Examination of Prevalence, Risk Factors, Diagnosis, and Treatment"

_nutrients, 2023, doi:10.3390/nu15173851_

Round 1
Reviewer 1 Report
This review article comprehensively analyzed orthorexia and psychiatric orthorexia, covering prevalence, risk factors, diagnosis and treatment strategies.
A major strength of the paper is the distinction between orthorexia and orthorexia nervosa. Pointing out these differences is important for the potential reader. In the section on the description of dignostic tools, I found a broader description of them lacking. The tools described basically measure various factors associated with orthorexia, which would be worth highlighting and relating to the distinction between orthorexia and orthorexia nervosa
In addition, the article sheds light on the growing concerns associated with these emerging eating disorders. The authors point out the complexity of diagnosing orthorexia and eating disorder due to the lack of standardized criteria and the need for further research to develop accurate assessment tools.
Author Response
Reviewer 1:
In the section on the description of dignostic tools, I found a broader description of them lacking. The tools described basically measure various factors associated with orthorexia, which would be worth highlighting and relating to the distinction between orthorexia and orthorexia nervosa
We thank the Reviewer for the comment. We added text on pages 8-9, lines 421-433 to clarify this issue:
Yet again, the differentiation between "orthorexia" and "orthorexia nervosa" remains a complex and debated issue within the field of eating disorders. The lack of a clear distinction in existing diagnostic tools poses challenges for accurate assessment and treatment. While "orthorexia" generally refers to an obsession with healthy eating and a fixation on consuming pure and uncontaminated foods [14], "orthorexia nervosa" suggests a more severe form involving distress, impairment, and potential physical or psychological consequences [1]. However, the absence of standardized diagnostic criteria for either condition makes it difficult to differentiate between them in practice. The concept of orthorexia nervosa is not yet formally recognized by major diagnostic manuals like the DSM-5, further complicating matters. Although clinicians and researchers are working to refine and establish clear criteria to identify and address these conditions effectively, the lack of consensus remains a significant hurdle in accurately distinguishing between them.
Reviewer 2 Report
This paper comprehensively reviews the emerging eating disorders of orthorexia nervosa and orthorexia, including epidemiology, risk factors, diagnostic criteria, and treatment approaches. The overall structure of the paper is logical, the arguments are clear, and the references are sufficient. However, there are a few aspects that could be further improved:
Title: Since there is currently no consensus on the diagnostic criteria, using "orthorexia nervosa" in the title may be misleading. I suggest modifying it to "orthorexia and orthorexia nervosa" to more accurately reflect the content.
Abstract: The distinction between orthorexia and orthorexia nervosa could be further emphasized, as well as the limitation of lack of standardized diagnostic criteria currently.
Keywords: Consider adding keywords like "risk factors" to more fully encompass the core contents of the paper.
Introduction: The analysis of the recent prevalence of "clean eating" ideals in society as historical background could be supplemented, as this sociocultural factor is closely related to the rise of orthorexia.
Results: In the risk factors section, I suggest adding a discussion of personal illness fears as a psychological factor. In the complications section, the analysis of impaired social functioning could be appropriately expanded.
Conclusion: The conclusion could further emphasize the urgent need to establish unified diagnostic criteria, which is an important direction for current research.
Overall, this paper comprehensively reviews the emerging topic of orthorexia, which is meaningful in promoting understanding of this disorder. The above are my review comments, which I hope can assist the authors in improving the paper. Please consider revising and supplementing based on the comments to enhance the academic value of the paper. I sincerely appreciate the authors' research work.
Author Response
Reviewer 2:
Title: Since there is currently no consensus on the diagnostic criteria, using "orthorexia nervosa" in the title may be misleading. I suggest modifying it to "orthorexia and orthorexia nervosa" to more accurately reflect the content.
We agree with the reviewer’s comment. The title was changed accordingly (page 1, line 2):
Orthorexia and Orthorexia Nervosa: A Comprehensive Examination of Prevalence, Risk Factors, Diagnosis, and Treatment
Abstract: The distinction between orthorexia and orthorexia nervosa could be further emphasized, as well as the limitation of lack of standardized diagnostic criteria currently.
A text was added to the abstract, page 1, lines 17-21:
The distinction between "orthorexia" and "orthorexia nervosa" is a debated issue in eating disorder research due to a lack of clear diagnostic criteria, making it challenging to accurately differentiate between an obsession with healthy eating and a more severe form with potential distress and impairment.
Keywords: Consider adding keywords like "risk factors" to more fully encompass the core contents of the paper.
The term “risk factors” is indeed found in the Keywords.
Introduction: The analysis of the recent prevalence of "clean eating" ideals in society as historical background could be supplemented, as this sociocultural factor is closely related to the rise of orthorexia.
This is an insightful comment raised by the reviewer. A text was added to the MS on page 2, lines 81-90:
In recent years, there has been a noticeable upsurge in the prevalence of "clean eating" ideals within society [28]. This trend reflects a growing societal emphasis on pursuing healthier lifestyles and making more mindful dietary choices. "Clean eating" generally entails prioritizing whole, unprocessed foods while limiting or avoiding heavily refined and artificial ingredients. This movement has fueled a desire for improved well-being, weight management, and increased energy levels [28]. However, discussions around clean eating also come with debates about the potential for creating rigid eating patterns, promoting unrealistic body standards, and contributing to the stigmatization of certain foods [2]. As this trend continues to evolve, striking a balance between informed nutritional choices and avoiding extreme dietary restrictions remains an ongoing conversation.
Results: In the risk factors section, I suggest adding a discussion of personal illness fears as a psychological factor. In the complications section, the analysis of impaired social functioning could be appropriately expanded.
We accept the reviewer’s comment and added the following text on page 4, lines 182-190:
The obsession with consuming only "pure" and "clean" foods has ignited discussions about the intersection of personal illness fears and extreme dietary habits. Individuals with orthorexia often feel overwhelming anxiety about consuming anything perceived as unhealthy, so their fixation on food quality can severely impact their physical and mental well-being [1]. This highlights the complex interplay between a genuine concern for health and the development of an unhealthy preoccupation with food choices. While originating from genuine concern, the fear of falling ill can morph into a consuming obsession that requires careful consideration within the broader context of mental health and balanced nutrition.
We also added the following text on page 6, lines 306-318:
Impaired social functioning is a salient aspect of orthorexia nervosa [32], [35]. As individuals with orthorexia become increasingly preoccupied with adhering to rigid dietary rules, their social interactions often suffer [1]. This is primarily due to the restrictive nature of their eating habits, which can lead to avoidance of social gatherings involving food, dining out, or participating in shared meals [33]. The intense anxiety surrounding food quality and composition may distance them from friends and family who do not share their dietary restrictions, resulting in isolation and alienation [2], [11], [25]. The rigid mindset and time-consuming behaviors associated with orthorexia can also hinder the ability to engage in spontaneous social activities, further compromising their social connections. Ultimately, impaired social functioning underscores the need for a comprehensive approach to address both the physical and psychological aspects of orthorexia nervosa to help individuals regain a healthy relationship with food and interpersonal interactions.
Conclusion: The conclusion could further emphasize the urgent need to establish unified diagnostic criteria, which is an important direction for current research.
We thank the reviewer for this comment and added the following text on page 13, lines 567-569:
Even though a recent consensus paper by Donini et al. [30] has made good progress toward a better understanding of the diagnosis and conceptualization of Orthorexia Nervosa, the establishment of unified diagnostic criteria is urgently needed to facilitate accurate identification, research, and effective treatment of this complex eating disorder.